# “I Carry the Trauma and Can Vividly Remember”: Mental Health Impacts of the COVID-19 Pandemic on Frontline Health Care Workers in South Africa

**DOI:** 10.3390/ijerph20032365

**Published:** 2023-01-29

**Authors:** Pinky Mahlangu, Yandisa Sikweyiya, Andrew Gibbs, Nwabisa Shai, Mercilene Machisa

**Affiliations:** 1Gender and Health Research Unit, South African Medical Research Council, Pretoria 0001, South Africa; 2Faculty of Health Sciences, School of Public Health, University of Witwatersrand, Johannesburg 2193, South Africa; 3Department of Psychology, Faculty of Health and Life Sciences, University of Exeter, Exeter EX2 4QG, UK; 4Institute of Global Health, University College London, London WC1E 6BT, UK; 5Centre for Rural Health, University of KwaZulu-Natal, Durban 4041, South Africa

**Keywords:** mental health, COVID-19 impacts, frontline health care workers, South Africa, pandemic

## Abstract

We know from research that pandemics and disease outbreaks expose HCWs to an increased risk of short and long-term psychosocial and occupational impacts. We conducted qualitative research among 44 frontline health care workers (FHCWs) practicing in seven South African hospitals and clinics. FHCWs were interviewed on their experiences of working during the first-wave of the COVID-19 pandemic and its perceived impact on their wellness. In this study, FHCWs included the non-medical and medical professionals in direct contact with COVID-19 patients, providing health care and treatment services during the COVID-19 pandemic. Most of the FHCWs reported stressful and traumatic experiences relating to being exposed to a deadly virus and working in an emotionally taxing environment. They reported depression, anxiety, traumatic stress symptoms, demoralization, sleep difficulties, poor functioning, increased irritability and fear of being infected or dying from COVID-19. The mental health impacts of COVID-19 on HCWs were also associated with increased poor physical wellbeing, including fatigue, burnout, headache, and chest-pains. FHCWs reported professional commitment and their faith as critical intrinsic motivators that fostered adaptive coping while working on the frontline during the first-wave of the COVID-19 pandemic. Many alluded to gaps in workplace psychosocial support which they perceived as crucial for coping mentally. The findings point to a need to prioritize interventions to promote mental wellness among FHCWs to ensure the delivery of quality healthcare to patients during pandemics or deadly disease outbreaks.

## 1. Introduction

As pandemics and infectious disease outbreaks occur, they require immediate responses and healthcare workers (HCWs) are on the frontline delivering health care. Systematic reviews show pandemics and disease outbreaks are associated with detrimental effects on frontline health care workers (FHCWs) [1,2]. For instance, during the Severe Acute Respiratory Syndrome (SARS) 2003 outbreak, many HCWs acquired SARS [3]. Moreover, FHCWs experienced negative psychological and occupational impacts, which endured even after the SARS pandemic had abated [4,5]. FHCWs worried about their personal safety and had anxiety about infection and transmission to family members, friends and colleagues [6]. Some reported that caring for colleagues was emotionally difficult and they experienced stigmatization and resentment about being chosen to care for patients with SARS [7]. FHCWs further reported burnout, fatigue, insomnia, irritability and decreased appetite during the SARS outbreak [2,8]. The increased workload, concern about the high risk of exposure and social stigma during the last SARS outbreak increased stress levels amongst HCWs [4,9].

The growing global literature shows that HCWs experienced high levels of stress, anxiety and depression during the COVID-19 pandemic [10,11]. A review and meta-analysis found that amongst HCWs, 30.0% experienced anxiety, 31.1% depression, 56.5% acute stress, 20.2% had potential post-traumatic stress and 44.0% experienced sleep disorders during the COVID-19 pandemic [12]. Similar findings were documented in other cross-sectional studies reporting distress, depression, anxiety and insomnia relating to the COVID-19 pandemic amongst HCWs [13,14]. Fear of the unknown or becoming infected were the primary concerns amongst HCWs [15]. HCWs had to work long-hour shifts under stressful conditions, which included having limited or lack of personal protective equipment (PPE), and experienced aggression from other medical staff during the COVID-19 pandemic [10]. Nurses, female workers, younger medical staff, and workers in areas with higher infection rates reported more severe degrees of all psychological symptoms than other HCWs [16]. Having a positive attitude towards the pandemic, social support and avoidance strategies protected against psychological distress [17]. Longitudinal studies demonstrate the enduring mental health impacts of COVID-19 amongst health care workers overtime [18,19].

Currently, there are few primary qualitative research studies to understand the impact of COVID-19 amongst HCWs in low and middle-income countries. The majority of studies have been conducted in Asia and high-income countries [11,16,20]. This paper aims to contribute to the literature from the global South on how FHCWs were impacted by COVID-19 while on the frontline during the first wave of the COVID-19 outbreak in South Africa. The findings of the study are essential for informing current policy and practice and to better prepare for future pandemics and infectious disease outbreaks.

## 2. Methods

### 2.1. Study Design

We conducted an exploratory qualitative study to explore the impact of COVID-19 on HCWs who were on the frontline providing health care services to COVID-19 patients during the first wave of the COVID-19 pandemic (7 June–22 August 2020); defined as the period from when COVID-19 weekly incidence was equal to or greater than 30 cases per 100,000 persons, until the weekly incidence was equal to or below 30 cases per 100,000 persons [21]. The study was conducted amongst HCWs in two provinces of South Africa, a middle-income country, one urban (Gauteng) and one rural province (Eastern Cape). Gauteng province had the highest number of confirmed COVID-19 cases during the 1st wave of the pandemic (206,892), followed by Kwa-Zulu Natal (110,521), Western Cape (104,781) and Eastern Cape (85,311) [22]. We purposively included provincial tertiary hospitals that were designated by the national government to provide care to COVID-19 cases and primary health care clinics that were referring patients to the selected tertiary hospitals. This was carried out in order to explore differences if any, regarding the impact of COVID-19 amongst FHCWs in a tertiary hospital and a local clinic. We selected two hospitals and two clinics in Gauteng province, and two hospitals and one clinic in the Eastern Cape province based on their availability and willingness to participate in the study and conducted in-depth interviews (IDIs) with FHCWs.

### 2.2. Data Collection

Data were collected between February and September 2021 from 44 FHCWs. We conducted IDIs with 25 FHCWs in Gauteng province, and 19 FHCWs in the Eastern Cape province working in COVID-19 wards, including high care and intensive care units. Recruitment adverts were posted in key areas accessed by FHCWs in hospitals and clinics, including reception, the COVID-19 ward and ICU entrances, tea rooms and cafeterias. Furthermore, hospital and clinic managers were requested and agreed to post the recruitment advert on the WhatsApp pages used as a communication channel with their teams in hospitals and clinics. The managers were also requested to make announcements about the study during meetings, which are held before teams start their shifts. The recruitment advert had a contact number and email address which FHCWs used to indicate their interest to participate in the study. The researcher would then follow-up to screen participants for eligibility and to arrange suitable time for the interview. In the screening process, FHCWs were specifically asked whether they were in direct contact and provided health care and treatment to COVID-19 patients during the first wave of the COVID-19 pandemic. Face-to-face recruitment was also conducted outside the COVID-19 wards, where researchers were given approval by hospital and clinic management to recruit participants for face-to-face interviews. All HCWs who were on the frontline during the first wave of COVID-19 had direct contact and provided health care and treatment to COVID-19 patients were eligible to participate. While we could not ascertain the percentage of those who volunteered to participate compared to the overall population of FHCWs in hospitals and clinics, we interviewed everyone who volunteered and was eligible to participate in the study. Table 1 describes the gender and designation of HCWs who participated in the study. Thirty-six interviews were conducted face-to-face and eight were conducted telephonically. Telephone interviews were conducted at the start of data collection as a safety measure to protect researchers and participants from the risk of acquiring COVID-19 when new cases were high. Face-to-face interviews were conducted after exploring various strategies and failure to recruit HCWs for telephone interviews. Face-to-face interviews were conducted between July and September 2021 when the number of new COVID-19 cases were less and the national vaccination programme for health workers was quite advanced. Physical distancing, washing hands, sanitizing and wearing masks were common protective practices in the population, as vaccination uptake was still lower than needed.

A semi-structured interview guide (Appendix A) with open ended questions was used to conduct interviews by the authors. We asked HCWs to describe their experience of working as a FHCW during the first wave of the COVID-19 pandemic and the impact it has had on their lives. We explored whether and how being on the frontline during the pandemic influenced dynamics in their homes, including relationships with partners and children. We further asked HCWs to describe the perceived impact of COVID-19 on their wellbeing. We asked them about their coping strategies and the support they received during the first wave of the COVID-19 pandemic. Most interviews, which were audio-recorded, were conducted in English; however, a few participants used a combination of English and vernacular, including IsiXhosa, IsiZulu, Setswana and TshiVenda. The interviews lasted between 30 and 45 min. Ethics approval for the study was granted by the South African Medical Research Council’s Human Research Ethics Committee (EC008-5/2020) and the University of Pretoria, Faculty of Health Sciences Research Ethics Committee (630/2020). Permission to conduct the study was granted by the Eastern Cape and the Gauteng Provincial Departments of Health, Hospital and clinic management. All participants provided written informed consent prior to participation in the interviews and were given R100 (7.13 USD) as a token of appreciation for their participation.

### 2.3. Data Analysis

Audio-recorded interviews from FHCWs were transcribed verbatim and those in other languages were translated into English by a research assistant. Data were analysed inductively using thematic analysis [23]. We specifically focused on FHCWs experiences of being on the frontline during the first wave of COVID-19 pandemic, and the impact it had on them. Themes were generated through the analysis process by PM and YS. Where there was disagreement in defining the themes, a third person (MM) was involved to assist in the decision-making process. First, we read the transcripts repeatedly to familiarize ourselves with the content of the transcripts and manually developed initial codes using MS Word. The initial codes were based on the interview guide and phrases representing segments of the text in the transcript. Next, we developed a codebook using raw data from the transcripts and reviewed and tested the data, leading to an expansion of codes. Next, text which seemed to fit together was grouped under a specific code [23]. Similar codes were then grouped to create the themes that are reported in this paper [23].

## 3. Results

Of the total 44 FHCWs recruited, the majority, 35 (80%), were female, 30 (68%) were nurses, and 33 (75%) were based in hospitals. Three (7%) of the nurses interviewed were also managing operations in a COVID-19 ward (See Table 1). Five main themes emerged from the interviews with FHCWs on their experiences and impact of COVID-19 in the early days of the pandemic. These include experiences of working in the COVID-19 ward, impacts on mental health, impacts on physical health, intrinsic motivation, and institutional support provided to FHCWs during the COVID-19 pandemic (See Appendix A). While the experiences and impacts refer specifically to the first wave, we recognize that they might have changed (or not) in the other waves of the pandemic. 

### 3.1. Experiences of Working in the COVID-19 Ward

#### 3.1.1. Working under Pressure with No Time to Grieve the Loss of Colleagues

FHCWs described the extreme pressure that came with working in the COVID-19 ward. Some indicated that while the hours worked in the COVID-19 ward were fewer compared to the other wards, “the shift felt like forever”:

“*We only worked inside the COVID-19 ward for 6 h compared to a normal shift [12 h], but that 6 h would be under extreme pressure, we would not even have time to eat.*” (Professional nurse, Gauteng, hospital)

While the shifts in the COVID-19 ward were shorter, there was more pressure to provide care to patients amongst FHCWs compared to other HCWs in the hospitals:

“*In [Hospital 1] we were working tough in our shift but at least at 10 h00 we would eat and you can go to the toilet in between. But here in COVID-19 ward we are working six hours. Like in the PPEs it is hot and maybe you are in a mask and you can’t get outside, your bladder is full and maybe you are hungry, so it’s so hectic…Six hours feels like 12 h, and even 12 h is better… you can spend the whole six hours standing…admitting patients, resuscitating, incubating, and transferring, it is so hectic.*” (Professional nurse, Gauteng, hospital)

Others spoke about how because the wards were busy, it did not allow them time to make sense or meaning of the death of their colleagues. FHCWs expressed frustration that they had to quickly heal from the loss, move-on and provide care to the next patient. They shared that there was no time to grieve the loss of colleagues who passed on during the COVID-19 pandemic:

“*I think they would make the announcements in our meeting telling us so and so (naming the person) has passed on or you will see reports on the social group like a WhatsApp group of the ward. The department will also support or go to that family or you just give (monetary) contribution where you can and that’s where it ends. We had to keep moving, carry on and be resilient. You can’t be sad while more people are dying and needing your attention. You just have to move on.*” (Doctor, Gauteng, hospital)

#### 3.1.2. Complexity of Treating COVID-19 Patients and Feeling Less in Control

Many spoke about the complexities of treating COVID-19 patients and watching some of them die from a virus that made the patient desaturate fast, presenting with complications within a short period of time:

“*It is hectic, COVID patients just change suddenly because you will nurse a patient then suddenly the patient just gets desaturated [drop in blood oxygen level] and starting to be distressed. We start to resuscitate, and incubate and 5 min later the patient dies…Then you have to call the family and explain to them what happened.*” (Professional nurse, Gauteng, hospital)

Another participant reiterated the difficulty of treating COVID-19 as thus:

“*A patient would arrive able to walk by themselves, then the condition will get worse and they die… Also sometimes seeing the patient arriving not in bad condition, but the condition became worse while admitted and they die.*” (Professional nurse, Eastern Cape, hospital)

FHCWs explained that witnessing all the complications patients experienced due to COVID-19 infection and the high number of deaths made them “feel out of control of the situation and frustrated” by not being able to help as they wanted to. Others spoke about feeling they did not do enough to care for patients who died because they were short-staffed:

“*We don’t have enough ICU staff, and sometimes you see that patients die not because of the virus but because of the lack of care…when you are short staffed you can’t give care to everyone and then you would feel like you are not doing enough for the patients even though you want to but you can’t. That stays with you for the rest of your life*.” (Professional nurse, Gauteng, hospital)

“*You find that one professional nurse had to look after some fifty confirmed cases of positive patients with COVID, maybe we would have two nurses or one staff nurse, I was working alone particularly in night shifts, one staff nurse one enrolled nurse and two ENAs with four or five patients, looking after fifty patients so it was straining us and you could only do so much*.” (Professional nurse, Eastern Cape, hospital)

### 3.2. Anxiety and Fear of Getting Infected and Death

The majority of FHCWs spoke about feeling particularly terrified when the pandemic started and having to be on the frontline to provide care to COVID-19 patients. They appraised their risk of infection and dying from COVID-19 as high:

“*At the beginning of the COVID we [HCWs] were so scared… First of all, when they said there is COVID you remember around April, or I think March, and we had the first patient, we were very scared that we are going to die, and we were the first group to come. They [managers] had to send us here [ COVID-19 ward] so people didn’t want to get inside [the COVID-19 ward]. Even the doctors were standing there and we needed them to give us prescription but they didn’t want to go there, they didn’t want to die because most of the nurses don’t want to die, the professional nurses were dying and I think the statistics was around 100 and something [deaths]. So, we were very scared, and every time we go there [COVID-19 ward] we would tell ourselves that we are not coming back [alive]… we were so scared knowing that we will leave our families and parents behind*.” (Professional nurse, Gauteng, hospital)

“*Everybody was terrified. Everybody in the ward is crying and you don’t feel like you can continue, you realize that you can’t expect normal emotions in an abnormal situation and everybody feels the same*.” (Doctor, Gauteng, hospital).

For some FHCWs, their anxiety and fear of infection and death were exacerbated by not having personal protective equipment (PPE) to protect themselves from contracting the virus. Most were anxious about contracting the virus and infecting family members:

“*I can say during the first waves it was very difficult because it was the first time for us to hear about COVID-19 and we were scared I don’t want to lie, and there were no PPEs at work by that time. So, we were supposed to go and nurse the patient without protection and as I am staying with my daughter at home, I was worried that I wonder what will happen to her. But we had to nurse the patients with COVID-19 anyway because there was nothing we can do*.” (Professional nurse, Eastern Cape, hospital)

The COVID-19 pandemic led FHCWs to confront the possibility of their own death and to put in place measures to protect and continue providing for their families if they died:

“*I ended up calling my attorney to help me with that because I felt that, why me? I have a small child, so that means I am going to die, and do you remember last year [2020] what was happening, so it was emotionally draining, and I even thought of resigning, but I have kids so I can’t stay at home, I ended up going to join [name of life insurance] a life cover, I joined it because of the pandemic so that if I die at least my kids must be left with something*.” (Professional nurse, Gauteng, hospital)

Other FHCWs explained that their fear of death was partly due to a lack of understanding of the disease in the early days of the pandemic and exposure to the high number of COVID-19 deaths:

“*So initially it was a bit difficult to adjust because we feared the unknown [COVID-19], we were not sure what we are dealing with you know, there was still uncertainty about how we should work, what to do and how we should divide ourselves in the COVID ward and certain people should be exposed than others, but now it’s a general thing like every day I know that I am going to work in a structure and everybody knows what is expected out there than before, that structure was still established*.” (Medical registrar, internal medicine, Gauteng, hospital)

Those with other underlying health conditions felt that they were more likely to die from COVID-19. Thus, they were more reluctant to work in COVID-19 wards, but felt compelled to do so:

“*Being a health care worker and having to work with the COVID positive patients, I was afraid. Initially I didn’t know and how to work with the patients of COVID, I tried all the means of going to my doctor to write me a letter to tell my manager that I can’t work in the COVID ward because I was afraid that being a health care worker with co-morbidities, I’m going to be part of the statistics. I am going to die from this*.” (Operations Manager, Gauteng, hospital)

Others felt that their managers did not care about them during a critical time when they needed their support:

“*We felt like they dumped us here and nobody came here, and nobody cared about us even the government did not care about us, so that is why I am saying it was affecting emotionally and every time when we had to come here [hospital] we were not feeling okay… I was feeling sad, actually I was so scared I never thought I will survive especially because I have underlying disease, I thought this corona is going to take me*.” (Doctor, Gauteng, hospital)

### 3.3. Mental Health Impacts of COVID-19

#### 3.3.1. “I Carry the Trauma”: Distress of Seeing Patients “Dying like Flies” Everyday

FHCWs said they felt stressed and traumatized from losing many patients and seeing them die in greater numbers than usual:

“*I remember one time it was Monday, you know when you come from home you are feeling fresh and you start a week, when I get to this other door, I came across four corpses and then I turned around and went that side I got four and I was like there is no other door that I will use, so I had to get in. When I got in, there were more, then I was not okay... I don’t want to lie; I went to the toilet I cried*.” (Care services staff, Gauteng, hospital)

FHCWs described their experience of seeing the many deaths as emotionally and mentally challenging:

“*It was so stressful and emotional because you will see people dying… Yes, you see people dying like maybe four people a day and it was emotional, and it is not something that you are used to like sometimes even in the unit that I was working in, maybe in three months there is only one death… Everyday people are dying! Some are still young and some are old*.” (Professional nurse, Gauteng, hospital)

“*Mentally and emotionally, you get affected because it’s very painful when you talk with the person and it seems she is happy or he is happy like saying ‘sister within two days I’m going home’, the next thing they don’t go home. They go home as a corpse. You feel for their kids and their family members, put yourself in their shoes, imagine how they feel—because they usually call every day to check on their relatives*.” (Professional nurse, Gauteng, hospital)

FHCWs also spoke about the trauma of losing their colleagues. They narrated how they transferred their colleagues’ death to themselves:

“*It was very sad because it wasn’t just only patients. There were some people we knew ‘colleagues’ as our patients that got affected by COVID-19 and some of them died… there was a lot of stress and sadness around the hospital… it was very sad, very sad and it was depressing that we couldn’t even go there to pay our last respect*.” (Doctor, Easter Cape, hospital)

A similar sentiment was shared by another participant:

P: Yes, and the other day we nursed our colleague who was here and it was not easy and she died, it was during the first waveF: Oh, so how did you feel? What went through your mind?P: I was scared I do not want to lie, when I got home sometimes, I couldn’t sleep because I was thinking what will happen to me because I am also working in the high care, and because I saw her two weeks before she was fine but in a space of two weeks, she is gone just like that (Professional nurse, Gauteng, hospital)

#### 3.3.2. Symptoms of Poor Mental Health among FHCWs: “Not an Easy Journey”

All FHCWs shared how working during the first wave of the COVID-19 pandemic impacted their mental health. Our findings suggest overlapping symptoms of traumatic stress triggered by difficult experiences and events during the COVID-19 pandemic. Symptoms mentioned included having intrusive thoughts of the events, emotional withdrawal or detachment, sleep difficulties and flashbacks:

“*Having to break the news and explain what happened has been emotionally draining. When there is nothing else you can do as a doctor for the patient, you have to let their family know. I carry the trauma and can vividly remember cases and families of all patients I have lost, the sadness on their families’ faces, the heart-break*.” (Doctor, Gauteng, hospital)

“*We were like losing our minds after going through the first wave. I could not sleep, always exhausted and experienced burn out*.” (Professional nurse, Eastern Cape, hospital)

“*This thing of COVID-19, it has traumatized us…I am having those flashbacks of the COVID-19 ward when the numbers were above 50 in the ward. You have to run all over knowing very well that you are also exposing yourself, but you don’t have another option because these people need you more, so it was not easy*.” (Operations Manager, Eastern Cape, hospital)

### 3.4. Impact on Physical Health

Linked to the traumatic stress symptoms, FHCWs reported feeling irritable, physically unwell and burnt out. They described being fatigued and having diminished day to day functioning:

“*I was feeling tired and when you are tired you are going to get weak, we were feeling weak and on top of that when we experienced burn out… when you are stressed even amongst co-workers we were fighting over small things like you would ask someone “why are you taking this person”, “why are you doing this”, so those fights shows that you are burnout because you are tired, so we were fighting and even the doctors were fighting, if the doctor gives you orders we would argue with them*.” (Specialist in critical care, Gauteng, hospital)

Another participant described how she was physically impacted:

“*I started experiencing constant chest pains, I don’t know if it was this mask or what I was going through emotionally. I would have chest pains that is heavy here [point to middle of her chest]*.” (Professional nurse, Eastern Cape, hospital)

For one doctor, COVID-19 negatively impacted his interest in carrying out his job, stating that he did not look forward to coming to work during the first wave of COVID-19. Below, he describes both the emotional and physical exhaustion he experienced:

“*I’m a very optimistic person and I love my job. Before COVID-19 I would not want to do anything else. I get up in the morning and I’m really excited to go to work. All of this changed, and I would not want to come to work. I started to get exhausted, headache, and I started to get irritated, we had to attend to too many cases, it was hectic. The workload nearly tripled, there was both a physical strain and an emotional strain*.” (Doctor, Gauteng, hospital)

### 3.5. Intrinsic Motivation and Peer Support

Despite the challenging experiences and negative impacts on mental health due to COVID-19, most of FHCWs adopted a positive attitude towards their work and this helped buffer some of the negative impacts. They described their intrinsic motivations which included commitment to the profession and faith as factors that helped them to be resilient while being on the frontline during the first wave of the COVID-19 pandemic:

“*It’s about just reminding yourself that one day you took an oath and by the time you were taking an oath you did not choose that I will run away when the situation is like this. It was just about telling yourself, also it was about putting yourself in the shoes of someone who need you at this moment, you see*.” (Operations manager, Eastern Cape, hospital)

Others reflected on how their faith supported them in their everyday work on the COVID-19 ward:

“*I just drew my strength from God, every day I would ask God to help me, I would tell God that I can’t do this alone and he has put me out here for a reason, so that’s how I deal with it, I don’t know about others*.” (Professional nurse, Gauteng, hospital)

FHCWs also valued the peer support and encouragement they provided each other in the workplace:

“*We have supported each other to deal with the anxieties, we encouraged each other to say ‘okay guys we are here and we said we are coming to deal with this [COVID-19], so let’s fight it’*.” (Professional nurse, Eastern Cape, hospital)

### 3.6. Institutional Support Provided to FHCWs

FHCWs described the workplace support they received during the first wave of the COVID-19 pandemic, which they appreciated. In a hierarchical setting such as hospitals and clinics, small gestures such as being given sweets or being asked how they were made FHCWs feel cared for and supported by their managers:

“*Sometimes the supervision staff they are coming this side bringing sweets just to cheer us up and to hear about us and the challenges we experience. Talking to them helps*.” (Professional nurse, Gauteng, hospital)

Simple actions such as managers discussing issues with FHCWs made the FHCWs less stressed:

“*Managers talk to us about everything related to COVID-19, so we don’t have stress [laughing]*.” (Assistant nurse, Eastern Cape, hospital)

Others described the psychosocial support they received from psychologists and social workers and the support received from religious leaders who were providing prayers in hospitals at the time. They valued the safe spaces that were created so that they could discuss their fears and receive support:

“*We do have a support especially the pastors outside they came in here and pray for us. We had a psychologist whom we talked to about COVID-19 and everything… If I need counselling, I just have to call then they come*.” (Assistant nurse, Gauteng, hospital)

“*There were psychosocial services that we were given, there was a social worker, we are also having a phone number whereby everybody was allowed to take a phone and call the lady and talk to her*.” (Operations manager, Eastern Cape, hospital)

However, there were some who felt that the support provided by management to staff that were infected by COVID-19 and in isolation was inadequate. Others suggested that FHCWs would have appreciated monetary incentives for the additional risks posed by COVID-19 infection:

“*Emotional support I think that’s still lacking and I think that’s something they need to improve on. I found out that once somebody gets COVID and now needs to recover for 10 days there is no support offered to that person. There was not much support, rather, you just go to isolate at home, and when you come back you are expected to catch up all the shifts that you didn’t do when you had COVID*.” (Doctor, Gauteng, hospital)

“*After having committed yourself and risked death during COVID-19, government should have provided monetary incentives for HCWs who were on the frontline. That to us would have demonstrated support*.” (Professional nurse, Eastern Cape, hospital)

## 4. Discussion

In this study, we explored the experiences of FHCWs and the perceived impacts of being on the frontline in selected hospitals and clinics in South Africa during the first wave of COVID-19. Accounts from FHCWs on their experiences of being on the frontline during the early days of the COVID-19 pandemic in hospitals and clinics in South Africa suggest that they worked under difficult conditions. COVID-19 was described as a health condition that raised high levels of anxiety and fear of death amongst FHCWs. Exposure to high numbers of deaths of patients and colleagues during the COVID-19 pandemic made FHCWs fear for their own lives and that of their families. COVID-19 became a public health crisis that rendered FHCWs helpless, caring for patients whose condition desaturated within a short period of time, with very little one could do to change the situation. Many felt that they were working in an environment where there was a lack of empathy and support from management. Lacking PPE at a critical time during the first wave exacerbated the fear and increased risk of infection amongst FHCWs, who were constantly worried about infecting family members. Some lived with regret and guilt after losing patients and felt that they could have done more to save lives if they were not short-staffed.

In our study, FHCWs reported poor mental health experienced during the first wave of the COVID-19 pandemic in South Africa. This finding is similar to that found in various other countries reporting increased levels of depression, anxiety and stress amongst FHCWs during the COVID-19 pandemic [10,11]. In our data, the majority of FHCWs reported stress related to high-risk exposure and worry about infecting family members. Working long hours in an environment with limited PPE, sometimes with limited staff, and treating a complex virus which was rapidly changing the patient’s condition, and continuously changing treatment protocols, which increased uncertainty, further increased stress levels. Other studies have found that uncertainty impacts the quality of service provided and can affect the mental wellbeing of HCWs. HCWs who are uncertain about how the patient may respond to treatment or how the condition may progress are likely to experience stress and depression [24,25]. If the patient does not recover, HCWs are likely to blame themselves and feel they have let patients and families down [26]. Our findings suggest a need to proactively identify the support needed and provide training to help build HCWs’ self-efficacy when treating a virus or disease where little is known about the prognosis. A study conducted by Lee, Wilson [27] found that HCWs who received training for COVID-19 and infection control and emergency procedures had less psychological distress than those who had not received similar training.

Many FHCWs in our study experienced anxiety and fear of death and reported depression and trauma, while being pressured to move on to caring for the next patient during the COVID-19 pandemic. The pandemic presented with an unprecedented high number of deaths, not only of patients but also colleagues, yet FHCWs did not have the opportunity to process the situation, nor grieve the loss of their colleagues. Rather, FHCWs were confronted with an increased fear of losing their own lives. Our data further highlight the link between poor mental health and being physically unwell amongst FHCWs during the first wave of the COVID-19 pandemic. Participants in our study spoke about having experienced fatigue and burnout, which made them feel tired all the time. Others described having constant chest-pains and headache. A few reported increased irritabilities, which sometimes resulted in unnecessary arguments with colleagues. These findings suggest that the day-to-day functioning of FHCWs was negatively affected during the first wave of COVID-19. Previous studies conducted elsewhere have shown that physical exhaustion and emotional distress are associated with suboptimal patient care, inefficiencies, and has long-lasting effects on the physical health status of HCWs [28]. Findings from our study implicate the need to bolster continual and structured workplace psychosocial support to enhance frontline workers’ self- efficacy, resilient coping and agility when dealing with a life-threatening and infectious novel virus under rapidly changing work environments. It has been long acknowledged that prioritizing HCWs is a requisite for effective health sector responses to pandemics and disease outbreaks [29].

Most FHCWs in our study had to isolate from family and friends to protect and minimize risk of exposure to COVID-19, in case they are infected with the virus. The social isolation negatively affected FHCWs’ mental health. FHCWs became each other’s support system during the difficult and stressful times of caring for COVID-19 patients during the first wave of the pandemic. Additional to the support they provided to one another, keeping a positive attitude including revisiting their reasons for choosing health care as a profession became their intrinsic motivating factor during the difficult times. The commitment to the profession and willingness to help others was amongst the reasons that kept HCWs motivated during the pandemic. The psychosocial support afforded to FHCW during the first wave of the COVID-19 pandemic varied significantly across hospitals and clinics. Some FHCWs received and appreciated the institutional support they received, including workplace counselling and managers who were appreciative of their efforts. However, others complained about a lack of workplace psychosocial support and empathy from their managers, and the absence of monetary incentives for their exposure to additional risks whilst on the frontline during the first wave of the pandemic. Other studies found that FHCWs who received psychological counselling and felt supported while caring for COVID-19 patients were less likely to experience depression, anxiety and stress [30]. Similar to other studies, our data support the key role of hospital managers or leaders in motivating the performance and resilience of HCWs during a pandemic [26,31]. This finding reinforces the importance of team-focused healthcare leadership that promotes positive working environments and prioritizes a culture of caring and wellbeing during pandemics [32]. Our data further highlighted the need for an involved manager or leader that acknowledges effort and shows appreciation and creates a safe space for open communication with staff. Preparation or training for hospital and clinic managers and leaders is essential to empower them to provide the needed leadership and mental health support to HCWs during pandemics and disease outbreaks [27].

These findings point to the importance of both individual and institutional level mental health support and interventions. At an individual level, interventions that foster a positive attitude, build resilience, and foster positive workplace relationships and social support systems are necessary. A workplace environment that considers the concerns of the employees and has leadership that is capacitated to provide mental health support for HCWs is critical. Interventions in the workplace can include routine mental health support, counselling services that promote positive thinking, and continuous debriefing. Social support from family and friends is also critical for helping HCWs to cope during a pandemic.

This study has several limitations. First, this was a qualitative study which relied on personal accounts and perceived impact of COVID-19 pandemic as described by FHCWs. We were not able to deduce the prevalence of stress, depression, anxiety, and fear as the study was not designed to measure the impact of COVID-19 amongst health care workers on any scale. Second, our sample did not allow us to look at profession and gender differences in the analysis. We had fewer males and doctors compared to females and nurses or other professionals in the study. However, our sample distribution is reflective of the male-female ratio in South African health care facilities, where females are over-represented compared to males. Furthermore, we did not ask participants about their age and duration of work in their respective occupation, which could have allowed for a better description of participant’s characteristics. Third, there is a likelihood of recall bias in our study. Interviews were conducted 6–12 months after the first wave ended. As such, we cannot ascertain whether reflections solely reflect the experiences and impacts of the first wave and not the other waves during the COVID-19 pandemic. Furthermore, FHCWs described their experiences as being traumatic, therefore this may have affected their recall and recollection. Lastly, the research included FHCWs in two provinces and from selected hospitals and clinics, thus it cannot be generalized to all health care workers in South Africa. Even so, we did not see any differences between the two provinces included in our study.

## 5. Conclusions

The study showed that many FHCWs in South Africa perceived the first wave of the COVID-19 pandemic as stressful and traumatic. FHCWs reported negative impacts on mental and physical wellness. They reported depression, stress, anxiety and fear of infection and anticipated death. The availability and quality of workplace psychosocial and managerial support provided to FHCWs during the first wave of the COVID-19 pandemic was variable. Individual resilience and collegial support were key to facilitate coping during the pandemic. Although we reported the mental health impacts of the COVID-19 pandemic on FHCWs at the beginning of the pandemic, it is possible that these impacts could have protracted or heightened during subsequent waves and at the peak of the pandemic. Notably, as the pandemic continued, the reported risks and impacts on FHCWs may have lessened or increased dependent on other factors such as access to vaccines and transmissibility or mortality rates due to the different mutants and strains of the COVID-19 virus. Additionally, there is evidence of many COVID-19 survivors experiencing the physical and physiologic symptoms of “long COVID” that could have mental health and other impacts amongst FHCWs [33]. These considerations highlight the importance of longitudinal and quantitative studies focused on understanding the trajectories of the risks and mental impacts of pandemics on the health workforce [8]. Furthermore, these findings warrant the prioritization of workplace psychosocial and managerial support interventions to address both the immediate and possible long-term mental health impacts of the COVID-19 pandemic amongst FHCWs. Overall, we conclude that protecting mental health and providing support for HCWs is of paramount importance. If not prioritized, poor mental health may compromise day-to-day functioning and negatively impact patient care.

## Figures and Tables

**Table 1 ijerph-20-02365-t001:** Health care workers demographic characteristics (N = 44).

	Overall N = 44	Profession				Gender	
		Doctors N = 8 (18%)	Nurses N = 30 (68%)	* Care Services N = 3 (7%)	COVID Ward Managers N = 3 (7%)	Male N = 9 (20%)	Female N = 35 (80%)
**Province**							
Gauteng	25 (57%)	6 (75%)	16 (53%)	2 (67%)	1 (33%)	5 (56%)	20 (57%)
Eastern Cape	19 (43%)	2 (25%)	14 (47%)	1 (33%)	2 (67%)	4 (44%)	15 (43%)
**Facility**							
Hospital	33 (75%)	6 (86%)	22 (73%)	2 (67%)	3 (100%)	9 (100%)	24 (69%)
Clinic	11 (25%)	1 (14%)	8 (27%)	1 (33%)	0 (0%)	0 (0%)	11 (31%)

* Care services includes non-medical HCWs such as cleaners, those who were bathing patients and changing linen, and kitchen staff providing food all of whom had direct contact with COVID-19 patients while rendering the service.

## Data Availability

The authors have made the interview guide available as a Appendix A to this submission. Furthermore, data can be made available on request from the corresponding author due to ethical obligations.

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
