# Peer review of "“I Carry the Trauma and Can Vividly Remember”: Mental Health Impacts of the COVID-19 Pandemic on Frontline Health Care Workers in South Africa"

_ijerph, 2023, doi:10.3390/ijerph20032365_

Round 1

Reviewer 1 Report

This well-written manuscript describes a useful study of the mental health impact of COVID-19 on frontline healthcare workers in South Africa early in the pandemic.

The following are my mostly minor comments and suggestions:

1.  Introduction, Line 34:  The article does not define Frontline Healthcare Workers.   It would be useful to define either in the abstract or introduction. 

2.  Introduction, Line 45:  The SARS outbreak is described as "SARS influenza".  SARS is not influenza, so the word influenza needs to be deleted.

3.  Methods, Line 90:  How were participants recruited and selected in each location?  How many were invited to participate.  How many declined?  What was the overall participation rate?

4.  Table 1, Line 102;  The headings appear bolded, but pharmacist is not bolded.  Since the number in some cells is very small (e.g. only 1 pharmacist) it would be advisable to merge cells with only 1 participant with another category as the person may be identifiable.   What was the age range of the participants?  What was the range of duration of work in their respective occupations?

5.  Methods, Line 128:  Notes that there were two reviewers of the data.  What happened if the reviewers disagreed?  Typically, a third reviewer may be needed to resolve differences between the first two.

6.  Results, Line 149 onwards:  The direct quotes should be in quotation marks, to differentiate them from the summaries. 

7.  It might be useful to the reader if the authors included a supplemental table with the interview questions they used. 

Author Response

An incorrect attachment was loaded on the system and I struggled to correct it. Below are the final responses:

Reviewer 2 Report

Qualitative research is difficult to collect, analyse and make conclusions. Some small mistakes in your paper make me doubt if all data are correctly collected and analysed.

Some examples:

1. In the abstract different spelling (FHCW and FCHW)

2. In data collection:

- abbreviation IDI without explanation

- different numbers in text and table 1: text 24 of Gauteng province/ table 25

In qualitative research, it is difficult to present results in a compact way. They should support the messages you want to provide. In addition, the discussion is also too long as the messages you want to give are not sufficiently outlined

Reviewer 3 Report

From my point of view, I suggest making a table with the themes. This could help to read the manuscript. In addition, a table with the sample/participants characteristics. 

Congratulations to make this manuscript. 
